# Nanostructure of nickel-promoted indium oxide catalysts drives selectivity in CO$_2$ hydrogenation

Matthias S. Frei[1], Cecilia Mondelli [1], Rodrigo García-Muelas [2], Jordi Morales-Vidal [2], Michelle Philipp[1], Olga V. Safonova [3], Núria López [2], Joseph A. Stewart [4], Daniel Curulla Ferré[4] & Javier Pérez-Ramírez [1✉]

Metal promotion in heterogeneous catalysis requires nanoscale-precision architectures to attain maximized and durable benefits. Herein, we unravel the complex interplay between nanostructure and product selectivity of nickel-promoted In$_2$O$_3$ in CO$_2$ hydrogenation to methanol through in-depth characterization, theoretical simulations, and kinetic analyses. Up to 10 wt.% nickel, InNi$_3$ patches are formed on the oxide surface, which cannot activate CO$_2$ but boost methanol production supplying neutral hydrogen species. Since protons and hydrides generated on In$_2$O$_3$ drive methanol synthesis rather than the reverse water-gas shift but radicals foster both reactions, nickel-lean catalysts featuring nanometric alloy layers provide a favorable balance between charged and neutral hydrogen species. For nickel contents >10 wt.%, extended InNi$_3$ structures favor CO production and metallic nickel additionally present produces some methane. This study marks a step ahead towards green methanol synthesis and uncovers chemistry aspects of nickel that shall spark inspiration for other catalytic applications.

[1] Institute for Chemical and Bioengineering, Department of Chemistry and Applied Biosciences, ETH Zürich, Zürich, Switzerland. [2] Institute of Chemical Research of Catalonia (ICIQ), The Barcelona Institute of Science and Technology, Tarragona, Spain. [3] Paul Scherrer Institute, Villigen, Switzerland. [4] Total Research & Technology Feluy, Zone Industrielle Feluy C, Seneffe, Belgium. ✉email: jpr@chem.ethz.ch

In heterogeneous catalysis, numerous systems rely on metal promotion to maximize process throughput[1,2]. Since these additives might carry stand-alone activity for undesired reactions jeopardizing selectivity and/or contribute to catalyst deactivation in conventional forms such as supported nanoparticles, specific nanostructures need to be designed to stabilize metal speciations displaying tailored electronic and geometric properties that minimize drawbacks while preserving benefits[3–7]. This often is a challenging task since the uniform production and in-depth characterization of precise atom-resolved structures lie at the frontier of current technologies.

In the frame of mitigating global environmental changes and lessening our reliance on fossil feedstocks[8–11], indium oxide was introduced as a breakthrough catalyst for $CO_2$ hydrogenation to methanol[12], exhibiting extraordinary high selectivity and superior activity and stability when supported on monoclinic $ZrO_2$[13–15]. Mechanistic investigations indicated that vacancies formed at a specific surface lattice position mediate $CO_2$ activation and $H_2$ heterolytic splitting[16–19], the latter unlocking the preferential formation of methanol instead of CO via the reverse water-gas shift (RWGS) reaction[18]. Still, since heterolytic $H_2$ activation is energetically demanding and limits the methanol synthesis rate, promotion with various hydrogenation elements was explored[20–30].

Platinum and palladium nanoparticles were shown to boost catalyst performance by aiding $H_2$ splitting, thus fostering $CO_2$ hydrogenation and generating additional vacancies on $In_2O_3$, but led to inferior methanol selectivity due to intrinsic RWGS activity and substantial reduction-induced $In_2O_3$ sintering. These pitfalls were mitigated by anchoring low-nuclearity (ca. 3 atoms) clusters to the $In_2O_3$ lattice in the case of gold and palladium[29,31]. Ruthenium and cobalt, typical metals leading to methane, were successfully employed when the first was alloyed with indium and the second encapsulated with $In_xO_y$[23,28]. In view of these findings, the low-cost alternative nickel, also a prototypic methanation metal[32], could find effective application in $CO_2$ hydrogenation if a favorable structure is identified. A recent study showed that metallic nickel on $In_2O_3$ did not produce methane in $CO_2$-based methanol synthesis, but clear support to its segregation from the oxide was not provided[26]. Another work indicated that $CO_2$ hydrogenation on Ni catalysts can be driven to CO and methanol by doping with indium[33]. In methane dry reforming on $In_xNi_y$ catalysts[34,35], CO adsorption on nickel was fully suppressed upon doping with indium, suggesting their alloying. In $CO_2$ dry reforming on $InNi/SiO_2$, the surface of Ni–In alloy particles was progressively covered with $InO_xH_y$ upon use[36], as for cobalt-promoted $In_2O_3$. These radical deviations of the behavior of pure nickel from its characteristic chemistry call for a rationalization of the nanostructures underpinning them, in comparison to the other metal promoters.

Herein, the synergistic interaction of nickel with indium oxide in $CO_2$-based methanol synthesis was explored through a comprehensive experimental and theoretical program. The nickel speciation was studied contrasting the behavior of coprecipitated and impregnated catalysts upon testing under industrially relevant conditions, with electronic effects being further evaluated on nickel deposited onto In–Al mixed oxides with distinct indium content. The nickel content was varied on the better performing impregnated catalysts identifying a clear impact on the product selectivity. In situ spectroscopy and diffraction methods along with thermal, volumetric, and microscopy analyses uncovered the nanometric construction of the selective nickel-poor and the unselective nickel-rich samples. Density functional theory (DFT) sheds light on the unique structural rearrangements of nickel deposited on $In_2O_3$ and the reactivity of complementary promoted surfaces, which was linked to experimental kinetic

parameters. Alloying of nickel with $In_2O_3$ emerged as key to provide uncharged hydrogen atoms to active sites on $In_2O_3$ while curtailing the nickel-mediated detrimental methanation pathway. Overall, this work gathers a fundamental understanding of a relevant system for sustainable methanol production and unravels structural and electronic features at the basis of the tunable selectivity of nickel in $CO_2$ hydrogenation routes.

## Results

**Impact of nickel content and synthesis method on activity and selectivity.** Nickel was incorporated into $In_2O_3$ by dry impregnation (DI, 1–20 wt.%, coded as $x$Ni-$In_2O_3$, $x = 1$–20) and coprecipitation (CP, 1–2.5 wt.%) aiming at a deposition on the oxide surface and formation of solid solutions to appreciate the role of metals intermixing and of the nickel oxidation state and chemical environment. $N_2$ sorption and X-ray fluorescence spectroscopy (XRF) indicated that all catalysts featured sufficiently high surface areas and nominal metal contents were closely matched (Supplementary Table 1). Assessing the samples containing 1 wt.% of nickel in $CO_2$ hydrogenation evidenced a higher methanol space-time yield ($STY$) compared to pure $In_2O_3$ ($STY = 0.16$ $g_{MeOH}$ $h^{-1}$ $g_{cat}^{-1}$, Fig. 1a), which, after 65-h equilibration, remained at a ca. twofold higher value (0.34 $g_{MeOH}$ $h^{-1}$ $g_{cat}^{-1}$) for the DI catalyst and levelled to a 25% higher value (0.20 $g_{MeOH}$ $h^{-1}$ $g_{cat}^{-1}$) for the CP catalyst. The stable behavior of the former contrasts palladium-promoted $In_2O_3$ prepared through DI, which experienced fast deactivation (Supplementary Fig. 1)[31]. The inferior performance of the CP catalyst is likely due to the synthesis approach burying a substantial portion of the promoter within the bulk of $In_2O_3$, but the presence of more nickel added by this method did not yield materials

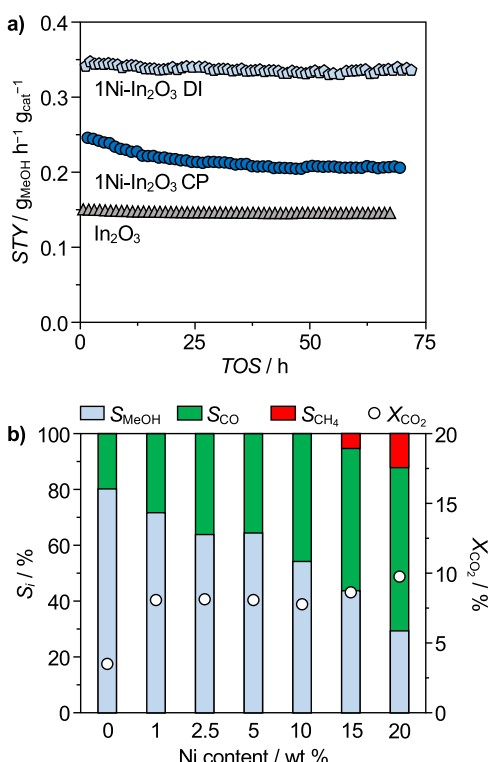

**Fig. 1 Catalytic performance of nickel-promoted $In_2O_3$ catalysts. a** Space-time yield ($STY$) of methanol as a function of time-on-stream ($TOS$) over Ni-$In_2O_3$ containing 1 wt.% of nickel incorporated by dry impregnation (DI) and coprecipitation (CP) and unpromoted $In_2O_3$ as a reference. **b** $CO_2$ conversion and product selectivity of catalysts produced by DI as a function of the nominal nickel content on $In_2O_3$. Reaction conditions: $T = 553$ K, $P = 5$ MPa, molar $H_2$:$CO_2 = 4$, and $WHSV = 24,000$ $cm_{STP}^3$ $h^{-1}$ $g_{cat}^{-1}$.

superior to those produced by DI (Supplementary Fig. 2). Considering the more favorable DI synthesis (Fig. 1b), methanol formation was progressively lowered in favor of the RWGS reaction up to a nickel content of 20 wt.%, while the $CO_2$ conversion was practically unchanged at ca. 8%, suggesting that sites activating $CO_2$ are unlikely located on the nickel phase. Methane formation was observed only at higher promoter contents ($S_{CH4} = 6$ and 13% at 15 and 20 wt.% nickel, respectively). This hints that $In_2O_3$ exerts a strong influence on the nickel properties and, only for high contents, a fraction of promoter remains unperturbed and can express its intrinsic behavior. To further address the selectivity switch, the amount of indium required to trigger it was explored by supporting 5 wt.% nickel on mixed indium-aluminum oxides with variable indium content (0–100 mol% In, Supplementary Fig. 3). While nickel on pure alumina was highly selective to methane ($S_{CH4} >$ 98%), 25 mol% of indium in the support sufficed to suppress methanation almost entirely ($S_{CH4} = 3$%). At the same time, the $CO_2$ conversion significantly dropped (from $X_{CO_2} = 18$ to 4%), corroborating that indium-modulated nickel species are significantly less active compared to pure nickel.

**Characterization of nickel 'speciation.** In-depth investigations were conducted to rationalize the behavior of the DI systems. Concerning catalyst reducibility, temperature-programmed reduction with hydrogen ($H_2$-TPR, Fig. 2a, Supplementary Table 2) evidenced that NiO conversion into metallic nickel occurs at ca. 340 K in all samples, preceding surface $In_2O_3$ reduction (370 K), which is substantially facilitated compared to the pure oxide (521 K). Based on the signal intensity in the profile of the 5Ni-$Al_2O_3$ reference, nickel was fully reduced in all cases. In 1Ni-$In_2O_3$, some $In_2O_3$ still reduced at its standard temperature, likely because the nickel amount was insufficient to facilitate reduction of the entire $In_2O_3$ surface. Diffuse reflectance infrared Fourier transform spectroscopy of adsorbed carbon monoxide (CO-DRIFTS) evidenced a weak signal specific to linearly bound CO ($2176\,cm^{-1}$) only for 10Ni-$In_2O_3$ and 15Ni-$In_2O_3$ (Fig. 2b), while pronounced bands of linear, bridged, ($2119\,cm^{-1}$), and three-fold ($2066\,cm^{-1}$) adsorbed CO was detected for the 5Ni-$Al_2O_3$ reference. Temperature-programmed desorption of carbon monoxide (CO-TPD, Fig. 2c) corroborated that the contribution of nickel to the adsorption of this molecule is significantly inferior when this metal is in contact with $In_2O_3$. These findings suggest remarkable electronic effects, i.e., metal-support interactions and/or alloying of nickel and indium[37], implying high dispersion, for the majority of nickel present, and small particle size for nickel unaffected by indium oxide adsorbing CO.

To shed further light on the promoter's features, catalysts were analyzed by additional methods. Scanning transmission electron microscopy coupled to energy-dispersive X-ray spectroscopy (STEM-EDX, Fig. 3a) revealed highly and almost homogeneously dispersed nickel in fresh lower-content specimens (1 and 5 wt.% Ni) and more agglomerated structures in fresh higher-content samples (15 and 20 wt.% Ni). Only in 20Ni-$In_2O_3$, some of the promoters appeared fully segregated from indium oxide. Slight nickel agglomeration is evident for all samples upon use in the reaction. Investigations by high-resolution transmission electron microscopy (HRTEM) of used materials did not visualize any nickel-based phases in 1Ni-$In_2O_3$ (Fig. 3b), although the magnification was chosen such that, based on the STEM-EDX results, some nickel must have been present within the imaged areas. In the 5 wt.% Ni sample, some amorphous islands can be found on $In_2O_3$, which might tentatively correspond to nickel-rich structures, due to their lower contrast compared to $In_2O_3$. In 15Ni-$In_2O_3$, a similar phase forms a ca. 1-nm thick layer covering many of the imaged $In_2O_3$ particles, which are additionally

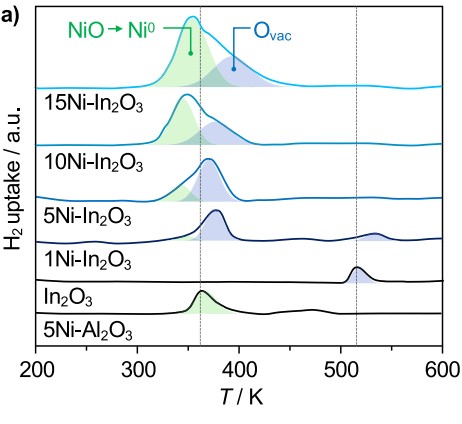

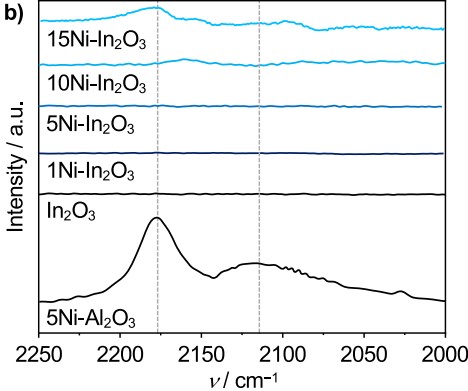

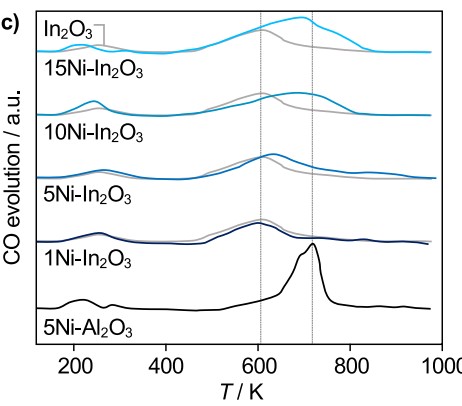

**Fig. 2 Sensitivity of nickel-promoted $In_2O_3$ catalysts to $H_2$ and CO. a** $H_2$-TPR at 5 MPa of pressure, **b** CO-DRIFTS, and **c** CO-TPD of Ni-$In_2O_3$ catalysts prepared by DI containing 1–15 wt.% nickel, with unpromoted $In_2O_3$ and 5Ni-$Al_2O_3$ serving as references. Deconvoluted signals specific to nickel reduction and oxygen vacancy formation on $In_2O_3$ are colored in green and blue in (**a**). a.u. = arbitrary units.

accompanied by poorly crystalline nickel-based particles in the 20 wt.% catalyst.

X-ray absorption near-edge structure spectra (XANES, Fig. 3c) of fresh 15 and 20Ni-$In_2O_3$ closely resemble that of $Ni^{2+}$ in NiO, while some discrepancies are observed for the lower-content solids, which can be explained by higher dispersion and stronger interaction with $In_2O_3$. In all but the highest nickel content catalyst, cationic nickel completely transformed into metallic species different from those in pure nickel metal upon use in the reaction, strongly suggesting alloying with indium. In 20Ni-$In_2O_3$, nickel is present in both metallic and oxidic states, likely because some NiO particles do not fully reduce due to their large size[32].

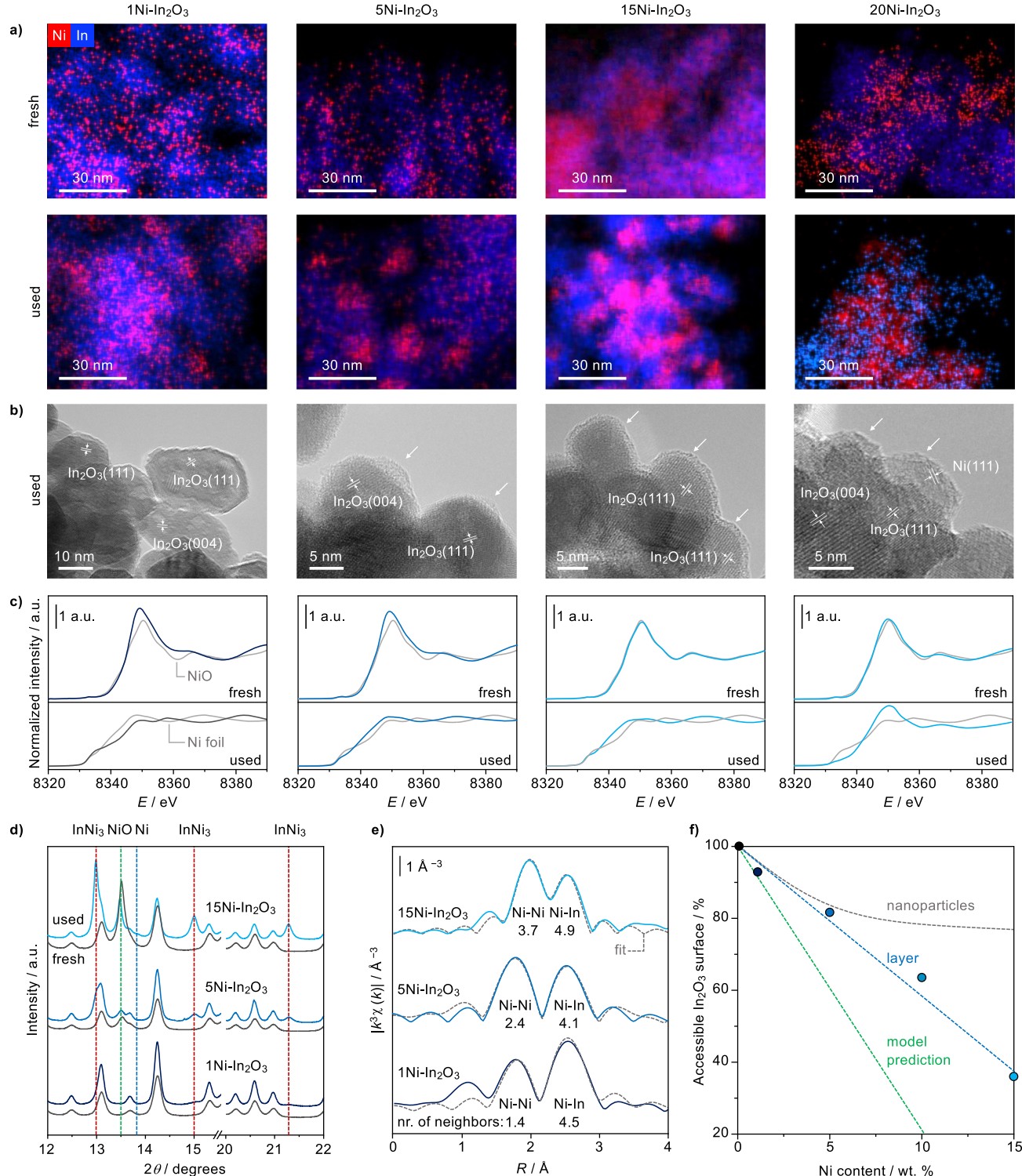

**Fig. 3 Structural and electronic elucidation of nickel-promoted In₂O₃ catalysts. a** STEM-EDX maps of Ni-In₂O₃ catalysts containing 1–20 wt.% Ni in fresh and used forms. **b** HRTEM images, with phases identified by fringe analysis and features of interest indicated, and **c** *k*-weighted Ni K-edge XANES, with spectra of NiO and metallic nickel serving as references, for the same used catalysts. **d** XRD patterns collected in monochromatic light and **e** EXAFS with model fit and an indication of neighbors' number along a specific scattering path for selected fresh and used samples. **f** Percentage of In₂O₃ surface not covered by nickel as a function of the nominal nickel content, determined by volumetric CO₂ chemisorption. The qualitative trend respective to nanoparticle formation and the coverage predicted by DFT is indicated. a.u. = arbitrary units.

High-resolution X-ray diffraction (XRD, Fig. 3d, Supplementary Fig. 4) measured in monochromatic light ($\lambda = 0.49292$ Å) indicated that NiO, present in the fresh materials, reduced indeed forming InNi₃ alloys. Their average crystal size is <1, 2.7 ± 0.7, and 7.3 ± 0.9 nm for the catalysts containing 1, 5, and 15 wt.%

nickel, respectively, in good agreement with the microscopy analyses. Alloy formation with indium shall be key to anchor the intermetallic phase strongly to the catalyst surface, preventing significant sintering. Only for 15Ni-In₂O₃, weak reflections specific to metallic nickel nanoparticles (ca. 1.2 nm) were also

detected, confirming the origin of methane production over this catalyst. Since this analysis necessitated catalyst operation directly within the capillary used for diffraction analysis and the consequent application of significantly lower flow rates than in the reactor, some NiO is still detected in the used solids. However, other characterization techniques mirroring the reaction environment more closely exclude that this phase exists upon reaction for these catalysts.

Analysis of the extended X-ray absorption fine structure (EXAFS, Fig. 3e, Supplementary Table 3) shows Ni–Ni and Ni–In bonds in the first coordination shell of nickel. The total number of first neighbors around nickel is 5.9, 6.5, and 8.6 for 1, 5, and 15Ni-In$_2$O$_3$, respectively. Since 12 neighbors are expected for nickel atoms in the bulk metal and the InNi$_3$ alloy, the alloy on all catalysts shall be well dispersed. Moreover, the number of Ni-Ni bonds progressively increases with higher nickel contents, i.e., 1.4, 2.4, and 3.7 for 1, 5, and 15Ni-In$_2$O$_3$, respectively. This is indicative of stronger nickel interaction with indium and suggests a two-dimensional layer-like morphology for the for the catalysts with lower nickel contentst, whereas particles additionally exist in higher-content samples.

Volumetric CO$_2$ chemisorption enabled to determine a linear decrease of the exposed In$_2$O$_3$ surface at increasing nickel content, consistent with a Stranski–Krastanov film growth (layer followed by nanoparticle formation)[38]. Indeed, an asymptotic decrease would be expected if nickel species formed agglomerates following a Volmer-Weber film growth[39] (Fig. 3f), as previously observed for the Pd-In$_2$O$_3$ system[31]. Although In$_2$O$_3$ should be fully covered at a nickel content of 12.5 wt.% (Supplementary Table 4), it is likely that inhomogeneous precursor distribution upon impregnation and the large anisotropy of the In$_2$O$_3$ surface prevented the formation of a perfectly uniform layer. Hence, ca. 40% of the In$_2$O$_3$ surface remained still exposed at a nickel content of 15 wt.%.

With a sound understanding of the DI systems, their structure was further investigated by first principles density functional theory. To represent the two-dimensional nickel phase, Ni(111) layers were placed on top of In$_2$O$_3$(111), identified previously and herein as the most abundant indium oxide termination (Supplementary Table 5)[18]. Based on the stability of nickel atoms with six neighbors each and to commensurate the oxide lattice (Supplementary Table 6), a slightly compressed nickel layer with 36 atoms per unit cell of In$_2$O$_3$ was considered more relevant than uncompressed layers with fewer nickel atoms (25–27). Interestingly, upon relaxation, nickel atoms in this discrete layer became more densely packed and extracted oxygen atoms from the underlying In$_2$O$_3$ (Fig. 4, Supplementary Movie 1). The driving force for this reconstruction can be traced back to the relative bond strengths, since Ni–In affinities are higher with respect to those of In-In and Ni-Ni (Supplementary Table 7), and is in line with other hydrogenation systems for which distinct but pronounced metal-support interactions were described, such as Cu-ZnO, Pt-TiO$_2$, Pt-CeO$_2$, and Pd-In$_2$O$_3$ (Supplementary Figs. 5 and 6, Supplementary Tables 8 and 9). The oxygen atoms extracted emerge to the outermost catalyst surface and are readily stripped as water by the hydrogen present in the reaction environment, which rationalizes the alloying of nickel with indium. Considering these dynamics, the oxygen atoms on the In$_2$O$_3$ termination were removed prior to placing the nickel layer. The most stable structure was found to be Ni$_{36}$-In$_2$O$_{3-\nu}$, which presents the maximal number of oxygen vacancies ($\nu$) on the oxide and nickel atoms (12 and 36, respectively, Supplementary Fig. 7). Still, since it cannot be excluded that a small fraction of the oxygen atoms that emerged at the outermost surface is dynamically stored on the InNi$_3$ patches upon CO$_2$ hydrogenation, the effect of residual four oxygen atoms at the most

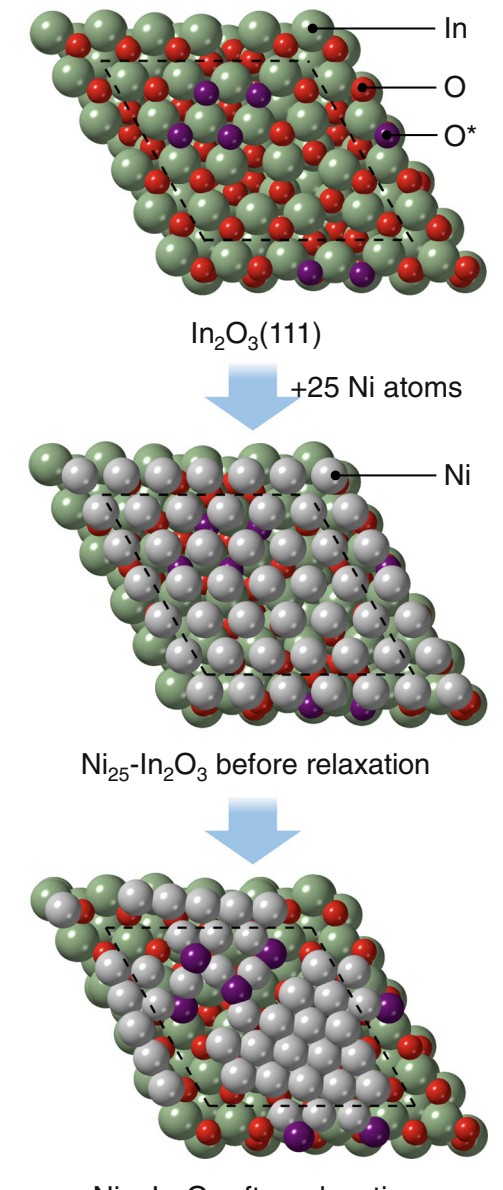

**Fig. 4 Reconstruction of an In$_2$O$_3$(111) unit cell bearing a metallic nickel layer.** The sketches show the restructuring of the interface between a nickel layer and the underlying In$_2$O$_3$. The pristine oxide surface is covered with a commensurate nickel layer, in this case containing 25 atoms. This structure is relaxed following a standard energy minimization algorithm. Given the corrugation of the In$_2$O$_3$ surface and the Ni-O and Ni–Ni relative binding energies, some oxygen atoms are stripped from the oxide (colored purple) and isolated hexagonally arranged Ni-patches are formed upon the relaxation. The dashed lines in the models indicate the border of the unit cell of In$_2$O$_3$ along with the (111) termination. The relaxation process is shown in Supplementary Movie 1.

stabilized positions was also explored (Ni$_{36}$O$_4$-In$_2$O$_{3-\nu}$). Finally, since the presence of isolated nickel species can also not be discarded, nickel atoms were alternatively deposited onto In$_2$O$_3$(111), i.e., Ni$_x$-In$_2$O$_3$, $x = 1$–4. These Ni$_x$-In$_2$O$_3$ structures are less stable than the nickel layer but more stable than NiO under a reducing atmosphere such as upon CO$_2$ hydrogenation. For instance, Ni$_2$-In$_2$O$_3$ is less stable than Ni$_{36}$-In$_2$O$_{3-\nu}$ by + 0.36 eV per nickel atom. Overall, the catalysts with up to 10 wt.% nickel are better represented by a combination of Ni$_{36}$-In$_2$O$_{3-\nu}$,

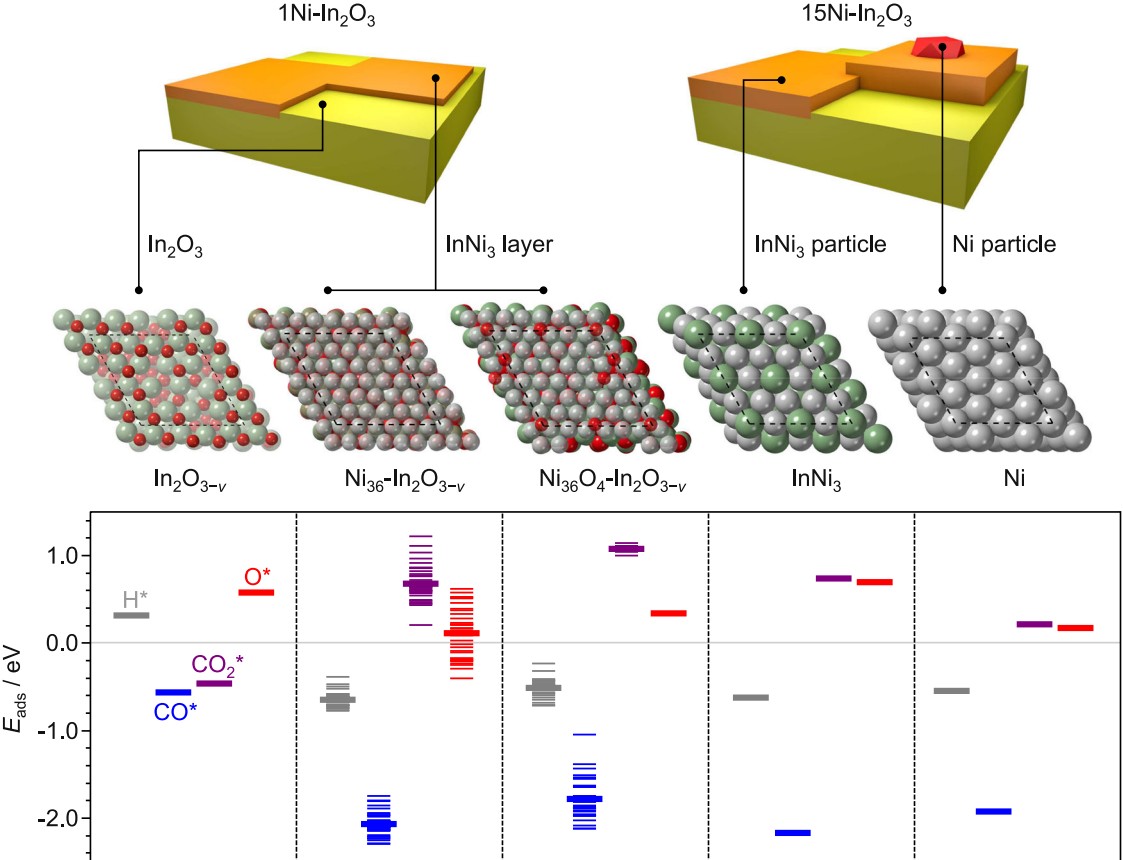

**Fig. 5 Adsorption energies of reaction species on nickel-promoted In₂O₃ catalysts.** Visualization of the structures of Ni-In₂O₃ catalysts with 1 and 15 wt.% nickel content loading (top), surfaces representative of the different catalyst constituents generated by DFT (middle), and adsorption energies ($E_{ads}$) of species relevant to the reaction associated with them (bottom). The multiple energy levels for individual species for models containing nickel layer relate to the presence of a distinct number of oxygen atoms. The subscript $v$ stands for vacancy. $CO_2$ adsorption at the boundary between unpromoted In₂O₃(111) and a strip of Ni(111) (layer equivalent to Ni₃₆-In₂O₃₋ᵥ) is shown in Supplementary Fig. 11.

InNi₃(111), and In₂O₃(111), while for higher-content materials InNi₃(111) and Ni(111) terminations, along with potentially Ni₂-In₂O₃ (vide infra), are additionally significant[40].

**Reaction mechanism and kinetics.** To explain the reactivity of the DI catalysts, the adsorption energies ($E_a$) of reactants on the distinct models were calculated (>200 adsorption calculations, Fig. 5). Ni(111) is associated with a mildly endothermic $CO_2$ adsorption, which can still be overcome (0.22 eV, Fig. 5), and barrierless hydrogen splitting and will readily transform carbon-based adsorbates into methane. In contrast, $CO_2$ adsorption is significantly weakened on clean InNi₃(111) (0.74 eV, Fig. 5), while hydrogen splits more exothermically and without an energy barrier. The chemical behavior of all layered nickel structures simulated is very similar to InNi₃(111) (Supplementary Fig. 8). Hence, any alloy type shall be virtually inert in converting $CO_2$ into any product on its own, but can provide hydrogen to the active site of In₂O₃(111) for coverages lower than one monolayer[18,31]. Nevertheless, the bulk alloy is expected to provide hydrogen radicals at a faster rate than the layer, due to its overall more metallic character. Even considering the presence of residual oxygen atoms upon catalyst operation, $CO_2$ adsorption remains far weaker on the nickel layers than on In₂O₃(111) or Ni(111). The structure retaining some oxygen atoms (Ni₃₆O₄-In₂O₃₋ᵥ) features inhibited CO adsorption compared to Ni₃₆-In₂O₃₋ᵥ. Considering the suppressed CO adsorption ability of the catalysts in CO-DRIFTS and CO-TPD analyses, it is conceivable that some

oxygen or hydroxide species populate the alloy film during reaction.

When considering low-content samples, the $CO_2$ hydrogenation performance can be seen as the convolution of that of a (multi)layer of InNi₃, pure In₂O₃, and nickel dimers on In₂O₃. The unpromoted In₃O₅ ensembles adsorb $CO_2$ and activate $H_2$ heterolytically[18]. Then, hydrides and protons are transferred to $CO_2$ forming methanol. On InNi₃(111) and Ni₃₆-In₂O₃₋ᵥ, $CO_2$ and CO adsorption are either endothermic or weaker than on In₂O₃, whereas homolytic $H_2$ adsorption is exothermic and $CO_2$ activated at the In₃O₅ ensemble on clean In₂O₃(111) shall be hydrogenated both with hydrogen split on the same active site and with hydrogen spilled from nickel layers[16], whereby sites at the periphery of the patches will be most relevant in the latter process (Supplementary Fig. 9). The fact that methanol selectivity is higher in nickel-poor catalysts than in nickel-rich ones suggests that hydrides and protons generated on In₂O₃ are still quite strongly utilized, since hydrogen radicals produced on the alloy favor both methanol and CO formation. In contrast to alloyed phases, low-nuclearity nickel clusters at the In₃O₅ ensemble, in particular Ni₂-In₂O₃, are expected to be highly active in the competing RWGS (Supplementary Figs. 10 and 11), in striking contrast to low-nuclearity palladium clusters anchored to the same ensemble.

To corroborate the DFT findings, kinetic analyses were carried out over all catalysts to experimentally assess the mechanistic origin of the promotional effect (Fig. 6a). The apparent activation energies for both methanol synthesis and the RWGS reaction,

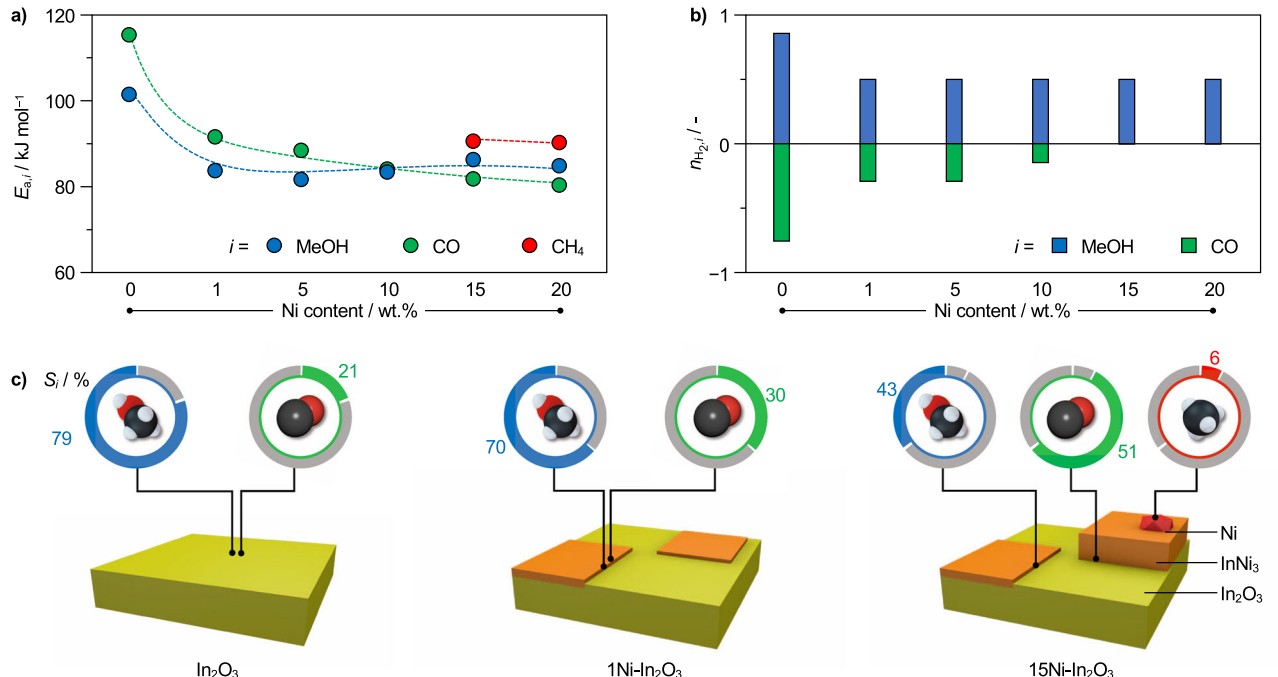

**Fig. 6 Nanostructure-driven kinetics and selectivity of nickel-promoted In$_2$O$_3$ catalysts. a** Apparent activation energies ($E_a$) and **b** reaction orders respective to H$_2$ ($n_{H_2}$) for methanol synthesis and the RWGS reaction over In$_2$O$_3$ catalysts as a function of their nickel content. **c** Graphical representation of the structures of In$_2$O$_3$ and Ni-In$_2$O$_3$ catalysts with low and high nickel loadings and indications where products shall be predominantly formed on them. The rings surrounding the molecules represent the respective product selectivity. Reaction conditions: $T = 553$ K, $P = 5$ MPa, $WHSV = 24{,}000$ cm$_{STP}^3$ h$^{-1}$ g$_{cat}^{-1}$.

extracted from catalytic tests conducted at variable temperature (Supplementary Fig. 12), are significantly lower already upon incorporation of the smallest nickel amount in comparison to pure indium oxide (from 101 to 83 and from 116 to 92 kJ mol$^{-1}$, respectively). The smaller difference between activation barriers for methanol and CO synthesis over 1Ni-In$_2$O$_3$ rationalizes its lower methanol selectivity compared to the unpromoted oxide (from 15 to 9 kJ mol$^{-1}$). A further increase of the nickel content has little impact on the activation energies, which reach values of 84 and 80 kJ mol$^{-1}$ for the RWGS and methanol synthesis for 20Ni-In$_2$O$_3$ respectively. The activation barriers for methanation on the materials active for this reaction (15 and 20Ni-In$_2$O$_3$) were determined at ca. 90 kJ mol$^{-1}$, in good agreement with literature on pure nickel catalysts (85–94 kJ mol$^{-1}$)[35,41,42] and corroborating the feasibility of methane formation on metallic nickel sites in these systems. Based on the weak dependence of the apparent activation energies for methanol and CO formation on the nickel content, the mechanisms leading to these products are likely highly similar throughout the materials. Hence, the progressive decay of methanol selectivity in favor of CO at higher loadings (Fig. 1b) has a kinetic origin. Since DFT calculations pointed to facilitated hydrogen activation as the origin of the promotional effect, apparent reaction orders respective to this reactant were determined from experiments at variable H$_2$ concentrations in the feed (Fig. 6b, Supplementary Fig. 13). For methanol synthesis, they decreased by equal amounts for all systems relative to bulk In$_2$O$_3$, from 0.8 to 0.5, in line with the stoichiometric coefficient of hydrogen splitting on the InNi$_3$ layers ($\frac{1}{2}$H$_2 \rightarrow$ H*). At higher partial pressures of H$_2$, the chemical potential of H* species adsorbed on the alloy layers increases as well, thus promoting H* spillover towards the In$_3$O$_5$ active site. The 0.5 reaction order suggests that the spillover mechanism dominates over the In$_3$O$_5$ on-site splitting at working conditions. For the RWGS reaction, the reaction orders increased from −0.7 to −0.3 for 1 and 5Ni-In$_2$O$_3$, i.e., the systems which contain mainly flat InNi$_3$ structures.

On these catalysts, both methanol and CO are produced on unpromoted In$_3$O$_5$ ensembles with the two paths competing for H* donated by the alloy patches. The negative reaction orders for the RWGS are explained by its first step (CO$_2$ + H* $\rightarrow$ COOH) being kinetically unfavored compared to the first of methanol production (CO$_2$ + H* $\rightarrow$ HCO$_2$, Supplementary Fig. 11a). The progressive, decrease in apparent activation energy for the RWGS reaction upon increasing Ni content might be explained based on the formation of additional metastable Ni$_1$-In$_2$O$_3$ and Ni$_2$-In$_2$O$_3$ ensembles selective to this competitive reaction (Supplementary Figs. 10 and 11). On Ni$_x$-In$_2$O$_3$ ensembles, H$_2$ splitting is barrierless, and thus the net reaction is not controlled by the partial pressure of H$_2$. At Ni contents of 15–20 wt.%, the RWGS reaction shall be mostly performed on these ensembles rather than on those free of nickel, and the reaction order with respect to H$_2$ decreases to zero. Overall, the kinetic data are in good agreement with earlier and above-presented investigations. Specifically, when H$_2$ is activated on pure In$_2$O$_3$, it is split into polarized species (H$^{\delta-}$ and H$^{\delta+}$) which are adsorbed on the In$_3$O$_5$ ensemble on an In$_3^{\delta+}$ substructure and O$^{\delta-}$ atom respectively[18]. The subsequent transfer to adsorbed CO$_2$ is energetically disfavored due to the strong polar interaction between adsorbed H$_2$ and the In$_2$O$_3$ surface. Neutral hydrogen atoms, provided by the alloy phases, do not have to overcome this energy barrier, thus leading to lower activation energies for methanol synthesis and the RWGS reaction[32]. However, consecutive proton and hydride supply to adsorbed CO$_2$ are highly selective towards methanol, whereas uncharged species foster both methanol and CO formation. Consequently, methanol synthesis cannot be enhanced beyond a certain threshold even if more homolytically split hydrogen is provided by more abundant alloy phases. On the contrary, the RWGS reaction is boosted to a greater extent in the presence of high nickel contents, presumably as a consequence of an excessive supply of hydrogen atoms (Fig. 6c).

## Discussion

Herein, the use of nickel as an economically attractive promoter for $In_2O_3$ in the direct hydrogenation of $CO_2$ to methanol was studied in fundamental and applied terms. Considering facile synthetic strategies, dry impregnation led to more stable and active catalysts than coprecipitation. Methanol synthesis was boosted along with the RWGS reaction to some extent and no methane was formed below a nickel content of 10 wt.%, despite the known high $CO_2$ methanation activity of nickel nanoparticles. In-depth characterization revealed a two-dimensional $InNi_3$ phase highly dispersed on $In_2O_3$ in nickel-lean samples, which is accompanied by nanoparticles of the same alloy as well as metallic nickel at progressively higher promoter contents. The formation of layered structures rather than agglomerated particles, due to peculiar wetting properties of nickel on $In_2O_3$ fostering film growth, and their strong anchoring on the oxide via alloying emerged as key contributors to the high catalyst stability. DFT simulations elucidated that indium-modulated nickel layers easily provide homolytically split hydrogen to $In_2O_3$, enhancing oxygen vacancy formation and contributing to $CO_2$ hydrogenation, while barely activating $CO_2$ on their own, which overall explains the beneficial effects and the lack of methane generation. Hydrogen radicals spilled from the $InNi_3$ phase can concomitantly support methanol and CO formation, while hydrides and protons produced on $In_2O_3$ preferably mediate methanol production. The variable relevance of the former species at distinct contents rationalizes the product distribution and kinetic parameters experimentally determined across all samples. The catalyst comprising 1 wt.% of nickel offers an optimal balance between charged and radical hydrogen atoms, reaching a doubled methanol STY compared to unpromoted indium oxide. Overall, this study identified key structural and electronic features controlling the performance of the classical hydrogenation metal nickel in contact with indium oxide relevant to attain a stable promoted system for a sustainable application. It also highlights that the atomic engineering of a promoter for $In_2O_3$ is strongly metal-specific, even when similarity in behavior is expected for elements belonging to the same group in the periodic table.

## Methods

**Catalyst preparation**. Unpromoted and nickel-promoted (1–2.5 wt.% nickel) $In_2O_3$ catalysts were prepared via a (co)precipitation (CP) synthesis similar to one reported earlier[31]. In addition, 1–20 wt.% and 5 wt.% nickel was added to pure indium oxide and to mixed indium-aluminum oxide supports with variable stoichiometry (0–100 mol% indium), respectively, by a dry impregnation (DI) method. The nickel-containing catalysts are labeled with the amount of promoter in wt.%, separated by a hyphen from the carrier, i.e., 5Ni-$In_2O_3$ indicates 5 wt.% nickel on $In_2O_3$. Details to all syntheses applied are provided in the Supplementary Methods.

**Catalyst characterization**. The metal content in the catalysts was determined by XRF, and porous properties of the catalysts were assessed by $N_2$ sorption. Catalyst reducibility was monitored by temperature-programmed reduction in hydrogen ($H_2$-TPR). CO adsorption was assessed by diffuse-reflectance Fourier transform infrared spectroscopy (CO-DRIFTS) and temperature-programmed desorption (CO-TPD). Nickel speciation, coordination geometry, and dispersion were accessed via X-ray absorption spectroscopy (XAS), X-ray diffraction in monochromatic light (XRD), STEM-EDX, and HRTEM. The surface area of $In_2O_3$ accessible to reactants was determined using volumetric chemisorption of $CO_2$. Details to all characterization techniques are available in the Supplementary Methods.

**Catalytic evaluation**. The experimental setup used for catalytic testing is described in detail elsewhere[18]. Briefly, all experiments were performed in a high-pressure continuous-flow fixed-bed reactor with an inner diameter of 2.1 mm surrounded by an electric furnace. In a typical experiment the reactor was loaded with 100 mg of catalyst with a particle size of 75–100 μm, which was held in place by a bed of quartz wool and heated from ambient temperature to 553 K (5 K min$^{-1}$) at 5 MPa under a flow of He (20 cm$^3_{STP}$ min$^{-1}$). After 3 h, the gas flow was switched to the reactants mixture (40 cm$^3_{STP}$ min$^{-1}$) comprising $H_2$ and $CO_2$ in a molar ratio of 4:1. To determine apparent activation energies, the reaction was initiated at 473 K and the temperature

stepwise increased to 553 K (increments of 20 K). Reaction orders with respect to $H_2$ were acquired applying a constant flow of $CO_2$ (8 cm$^3_{STP}$ min$^{-1}$) and increasing the flow of $H_2$ (from 20–32 cm$^3_{STP}$ min$^{-1}$, increments of 3 cm$^3_{STP}$ min$^{-1}$), while using He to balance the total flow to 40 cm$^3_{STP}$ min$^{-1}$. Ethane (0.5 cm$^3_{STP}$ min$^{-1}$, Messer, >99.9%) was added to the effluent stream to serve as an internal standard before the stream was sampled every 20 min and analyzed by online gas chromatography. The evaluation procedure of gas chromatography data is reported in the Supplementary Methods. Materials were tested for 16 h for performance comparison, catalyst stability was established over 72 h on stream, and, during kinetic tests, data were collected for 3 h at each condition and averaged. The absence of intra- and extraparticle diffusion limitation during kinetic tests were corroborated by the fulfillment of the Weisz-Prater and Carberry criteria.

**Computational methods**. DFT simulations were conducted with the Vienna ab initio simulation package (VASP) using the Perdew-Burke-Ernzerhof (PBE) density functional[43–45]. Core electrons were described by projector augmented-wave pseudopotentials (PAW)[46], while valence electrons were expanded from a plane-wave basis set with a kinetic energy cutoff of 500 eV and a reciprocal grid size narrower than 0.025 Å$^{-1}$. Bulk metal, intermetallic, and oxide structures relevant to investigate $In_2O_3$ promotion by nickel were modeled from their stable structures at ambient conditions. All bulk structures were fully relaxed and formation energies were obtained taking the bulk elements and gas-phase $O_2$ as reference. Spin-polarization was considered for Ni-containing systems.

The most abundant termination of bixbyite $In_2O_3$, the (111) surface[16,18], was modeled as a $p(1 \times 1)$ slab containing five O-In-O trilayers. The two outermost layers were allowed to relax and the three bottommost layers were fixed in their bulk positions. This surface is 14.56 Å wide, corrugated, and highly anisotropic. It features a protrusion, which is the active site for $CO_2$ hydrogenation to methanol. To represent $In_2O_3$ catalysts with low nickel contents, a nickel atom was adsorbed on the pristine $In_2O_3(111)$ surface between three oxygen atoms of the protrusion at symmetrically inequivalent positions. This process was repeated for low-nuclearity clusters containing 2–4 nickel atoms. Finally, 1–3 oxygen vacancies were created to check the ability of these clusters to favor oxygen abstraction. Besides, different nickel layers were accommodated onto $In_2O_3$ deriving from the $5 \times 5$, $3\sqrt{3} \times 3\sqrt{3}$, and $6 \times 6$ expansions of a Ni(111) monolayer, containing 25, 27, and 36 Ni atoms, respectively. Each layer was placed on $In_2O_3(111)$ surfaces with 0, 1, 2, 3, 6, 9, and 12 vacancies considering three different translations. To describe $In_2O_3$ catalysts with high nickel contents, the $InNi_3(111)$ and Ni(111) surfaces were also tested. The mechanism and energetics of $CO_2$ hydrogenation were investigated considering the adsorption of relevant species and full reaction paths[18,31]. Transition states were obtained from the climbing image nudged elastic band (CI-NEB)[47] and improved dimer method (IDM)[48]. Details to the calculations specific to surfaces containing low-nuclearity clusters and to metal-support interactions are provided in the Supplementary Methods.

## Data availability

The authors declare that the data supporting the findings of this study are available within the article and its Supplementary Information file. The DFT data are accessible at the ioChem-BD database at https://doi.org/10.19061/iochem-bd-1-183. All other relevant source data are available from the corresponding author upon reasonable request.

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

## Acknowledgements

Total Research & Technology Feluy is thanked for sponsoring this project. Dr. S. Mitchell and Dr. F. Krumeich are thanked for the electron microscopy measurements, and the Scientific Center for Optical and Electron Microscopy (ScopeM) at the ETH Zurich for the use of their facilities. We are grateful to Dr. Nicola Casati for performing the in situ XRD analyses. Dr. Marcos Rellán-Piñeiro is thanked for his input in the theoretical calculations. The Spanish Ministry of Science and Innovation RTI2018-101394-B-I00 project is acknowledged for financial support and the Barcelona Supercomputing Center – MareNostrum (BSC-RES) for providing generous computer resources.

## Author contributions

J.P.-R. and C.M. conceived and coordinated all stages of this research. M.F. and M.P. prepared and characterized the catalysts and conducted the catalytic tests. O.S. coordinated acquisition and performed the evaluation of X-ray absorption spectroscopy data. R.G.-M., J.M.-V., and N.L. conducted computational studies. J.A.S. and D.C.F. contributed setting industrial targets for the experimental program. All authors contributed to the writing of the manuscript.

## Competing interests

The authors declare no competing interests.
