## [Peer Review File · Nature Communications]

REVIEWER COMMENTS

Reviewer #1 (Remarks to the Author):

In this article, the authors reported very interesting metal-promoter effect and modulated selectivity by varying Ni content in Ni/In₂O₃ catalysts for the hydrogenation of CO₂. This work is surely important for converting CO₂ to value-added chemicals (e.g., methanol), for attacking the grand challenge of alleviating CO₂ emission. The authors found that impregnating small amount of Ni (1%) onto In₂O₃ can double the space time yield (STY) towards methanol production. Further increasing the loading amount of Ni (e.g., 10% and 20%) can cause the change of selectivity, leading to the production of more CO and CH₄. The authors have done an excellent job on analyzing the catalyst nanostructures of using XRD, H₂-TPR, CO-TPD, CO-DRIFTS, STEM-EDX, XANES, EXAFS, and DFT calculations, and identified the presence of layered Ni, NiIn₃ alloys, and Ni nanoparticles. The increased STY was explained by the facile production of hydrogen radicals on layered Ni, which are spilled into the In₂O₃ active site with bound CO₂ for the production of methanol. Reaction activation energies were deduced experimentally to rationalize the selectivity changes with the varying amount of Ni on In₂O₃. Sufficient experimental and computational data were provided to understand the metal promoter effect and the changing selectivity, and the manuscript is very well presented. I would recommend the manuscript to be published in Nature Communications, subjected to the following minor revision.

1. On Page, Line 17, please remove one “bound” in the sentence of “linearly bound bound CO (2176 cm⁻¹) only for 10Ni-In₂O₃ and 15Ni-In₂O₃”.
2. For the methanol production by 1Ni-In₂O₃, the STY increased and the activation energy decreased. The authors explained that the layered Ni structure can help to produce hydrogen radicals, which can lead to the increasing of STY as a result of increasing concentration of reactants (e.g., H radicals, H⁻, and H⁺). Please comment explicitly in the main text on what factors may have caused the decrease of activation energy.

Reviewer #2 (Remarks to the Author):

This is a contribution in the field of CO₂ reduction to methanol using Ni on indium oxide catalyst. Many techniques have been used to characterize this catalyst.

According to the results of characterization the active catalyst is composed of InNi₃ patches on the oxide of Indium.

The conclusions are based on the complex interpretation of these different techniques, including DFT.

I am personally reluctant to accept the paper for the following reason:

In₂O₃ has been reported to be a catalyst by itself in the reduction of CO₂ to methanol. reference 12
Pt, Pd and Co other metal supported on In₂O₃ have already been published (see the reference 20,21 23
24 ...

This paper just introduce Ni as a new metal promoted Indium oxide.
There is no real mechanism of CO₂ hydrogenation at a molecular level and no structure activity relationship

Reviewer #3 (Remarks to the Author):

Perez Ramirez and coworkers, in a cooperation between PSI, ICIQ, Total and ETH Zurich, have studied in great detail - as a follow up of their earlier efforts in this field of CO₂ activation - the promotion/effect of Ni on In₂O₃, a catalytically active phase for the transformation of CO₂ into CH₃OH. The authors, by using a combination of advanced experimental methods as well as theoretical calculations, determine the effect of Ni on the In₂O₃ as graphically illustrated in Figure 6c. It is clear that extremes can be noted when comparing Ni as metal phase (making mainly methane) and In₂O₃, which is very active and selective towards methanol. The authors explain this by the formation of different phases in the catalyst system, and also the alloy formation between Ni-In; and separate phases. I am overall positive about the work and recommend the publication for Nature Communications, although some revisions and additions are needed before the work can be published. These recommendations are outlined below:

1. The authors discuss the relative contributions of neutral and charged hydrogen species, but the whole idea of radical species is not much supported by experimental methods, including for example EPR. Hence, the discussion remains for me a bit superficial and could be strengthened/further elaborated. The result of this is that the entire discussion on how the catalytic system actually operates; and what the effect of Ni is still vague; not clear to the reader/me. I have tried to envisage it but not completely understood it. So, there is more information/backing needed to explain what is exactly happening.

2. The crux of the paper is the analysis of the XAFS spectra in combination with HRTEM); especially Figures 3b/c and the supporting information (Tables). Here, a more quantitative analysis, and extension to other catalyst materials are needed. Actually, I believe the paper would benefit much if the work was extended with a full analysis (including an expansion for Figure 6c to show next to the current samples also sample with 5 and 20 wt% Ni; hence where 5 samples are shown/discussed/analyzed in detail instead of the current 3 ones). Although I am aware that this requires more work it would much better focus on the different stages of Ni-In interaction. Now, some material is already some work included in the SI for the 5 wt% Ni sample, but more is clearly possible here, especially the PSI facility is close by. With this it would become more evident how an In-based methanol catalyst gradually changes in a Ni-based methane catalyst. Within this context I also refer to the use of IR probe molecules, which could differentiate between the alloying and different sites formed (see e.g. Nature Communications 2019, 10, 5330).

Summarising, overall this is very interesting and novel research work on an important research topic, which deserves publication in a major research journal. I recommend the work after proper revisions and extensions, as outlined above.

Manuscript NCOMMS-20-40209 - Response to Reviewers

Comments in *blue* - Replies in black - Actions in **bold**

Indicated page, line, or figure numbers refer to the revised manuscript and/or supplementary information with changes highlighted

Reviewer #1

In this article, the authors reported very interesting metal-promoter effect and modulated selectivity by varying Ni content in Ni/In₂O₃ catalysts for the hydrogenation of CO₂. This work is surely important for converting CO₂ to value-added chemicals (e.g., methanol), for attacking the grand challenge of alleviating CO₂ emission. The authors found that impregnating small amount of Ni (1%) onto In₂O₃ can double the space time yield (STY) towards methanol production. Further increasing the loading amount of Ni (e.g., 10% and 20%) can cause the change of selectivity, leading to the production of more CO and CH₄. The authors have done an excellent job on analyzing the catalyst nanostructures of using XRD, H₂-TPR, CO-TPD, CO-DRIFTS, STEM-EDX, XANES, EXAFS, and DFT calculations, and identified the presence of layered Ni, NiIn₃ alloys, and Ni nanoparticles. The increased STY was explained by the facile production of hydrogen radicals on layered Ni, which are spilled into the In₂O₃ active site with bound CO₂ for the production of methanol. Reaction activation energies were deduced experimentally to rationalize the selectivity changes with the varying amount of Ni on In₂O₃. Sufficient experimental and computational data were provided to understand the metal promoter effect and the changing selectivity, and the manuscript is very well presented. I would recommend the manuscript to be published in Nature Communications, subjected to the following minor revision.

We are pleased that the Reviewer recognized the significance and quality of our contribution and are grateful for his/her accurate assessment.

1. On Page, Line 17, please remove one “bound” in the sentence of “linearly bound bound CO (2176 cm⁻¹) only for 10Ni-In₂O₃ and 15Ni-In₂O₃”.

We have deleted the repeated word upon revision (page 6, line 20).

2. For the methanol production by 1Ni-In₂O₃, the STY increased and the activation energy decreased. The authors explained that the layered Ni structure can help to produce hydrogen radicals, which can lead to the increasing of STY as a result of increasing concentration of reactants (e.g., H radicals, H[•], and H⁺). Please comment explicitly in the main text on what factors may have caused the decrease of activation energy.

Upon activation on the In₃O₅ ensemble in pure In₂O₃, H₂ splits into polarized H^{δ-} and H^{δ+} species, which are adsorbed on an In₃^{δ+} substructure and an O^{δ-} atom, respectively, as detailed in refs. 18 and 31 of the previous version of our manuscript. The transfer of these species to adsorbed CO₂ is energetically demanding due to their strong polar interaction with the In₂O₃ surface. On the contrary, neutral radicals produced through H₂ activation on the Ni-containing phases are less stabilized by the oxide surface and can hydrogenate adsorbed CO₂ without an energy barrier. Thus, the activation energy for CO₂ hydrogenation with these hydrogen intermediates is lower than for the naked carrier alone. **We have detailed these aspects in the amended manuscript (page 14, line 2 to page 15, line 3).**

Reviewer #2

This is a contribution in the field of CO₂ reduction to methanol using Ni on indium oxide catalyst. Many techniques have been used to characterize this catalyst. According to the results of characterization the active catalyst is composed of InNi₃ patches on the oxide of Indium. The conclusions are based on the complex interpretation of these different techniques, including DFT. I am personally reluctant to accept the paper for the following reason:

We thank the Reviewer for appreciating the breadth of our experimental and theoretical efforts. Still, we are dismayed that s/he failed to acknowledge the novelty of our contribution, which does not lie in the introduction of a new metal promoter for In₂O₃, but on the rationalization of the unprecedented phenomenology in nickel-promoted CO₂ hydrogenation, as clearly recognized by the other Reviewers.

1. In₂O₃ has been reported to be a catalyst by itself in the reduction of CO₂ to methanol. Reference 12. Pt, Pd and Co other metal supported on In₂O₃ have already been published (see the reference 20,21 23 24 ...). This paper just introduce Ni as a new metal promoted Indium oxide.

This is an oversimplified description of the scope of our paper, which may reflect a lack of detailed reading by the Reviewer. Although a few metal promoters have already been studied, we would like to point out that vastly different catalyst design strategies are required to curtail their individual intrinsic activity for undesired competitive reactions while preserving beneficial attributes. In the case of cobalt, strong metal-support interactions enable encapsulation of the metal with In₂O₃, inhibiting CO₂ methanation (ref. 23 in the original manuscript). For platinum and palladium, charge transfer phenomena and stable atomic dispersion, respectively, are essential to suppress homolytic H₂ splitting, hindering the RWGS reaction (refs. 25 and 31). This study shows that, in the case of nickel, alloying with the oxide is a prerequisite to suppress CO₂ methanation, a so far undocumented phenomenon for an In₂O₃ promoter, which may be relevant for other catalysis applications. Hence, structural changes induced by the dopants span a very wide variety of behaviors requiring a proper systematization and our detailed experimental-computational characterization shows the way how these analyses should be conducted. **We have stressed this aspect better in the revised introduction of the manuscript (page 4, lines 21-23).**

2. There is no real mechanism of CO₂ hydrogenation at a molecular level and no structure activity relationship

We refute this statement. In fact, molecular-level understanding permeates our study, as firmly recognized by Reviewers #1 and #3. Our early work examined the CO₂ hydrogenation mechanism on In₂O₃ based on the supply of protons and hydrides (ref. 18) and described the impact of hydrogen radicals provided by metastable palladium nanoparticles (ref. 31). The former path is highly selective towards methanol, whereas both methanol and CO production are fostered in the latter. In nickel-promoted systems, the two mechanisms occur in parallel and an optimal promotional effect was attained by balancing their contributions through a specific catalyst architecture featuring patches of InNi₃ alloy well dispersed on the oxide. We thus disagree on the stipulated lack of structure-activity relationship. The contribution of diverse nickel phases arising at different nickel loadings has been elucidated in detail through the many techniques remarked the Reviewer. Moreover, the chemistry of all nickel species detected experimentally, including flat and tridimensional InNi₃ alloys and metallic nickel particles, has been studied through extended DFT simulations. Additional structures potentially stabilized by the oxide, *i.e.*, single atoms and low-nuclearity clusters at the active ensemble on In₂O₃, were also interrogated by theory providing a complete description of their reaction energetics. Finally, experimental findings and DFT predictions were linked *via* a kinetic analysis and summarized in Figure 6. Nevertheless, also in response to a suggestion by Reviewer #3, we have **deepened the characterization of the diverse nanostructures and further elaborated on their role in CO₂ hydrogenation (page 7, lines 6 to page 8 line 3; and page 14, line 2 to page 15, line 3).**

Reviewer #3

Perez Ramirez and coworkers, in a cooperation between PSI, ICIQ, Total and ETH Zurich, have studied in great detail - as a follow up of their earlier efforts in this field of CO₂ activation - the promotion/effect of Ni on In₂O₃, a catalytically active phase for the transformation of CO₂ into CH₃OH. The authors, by using a combination of advanced experimental methods as well as theoretical calculations, determine the effect of Ni on the In₂O₃ as graphically illustrated in Figure 6c. It is clear that extremes can be noted when comparing Ni as metal phase (making mainly methane) and In₂O₃, which is very active and selective towards methanol. The authors explain this by the formation of different phases in the catalyst system, and also the alloy formation between Ni-In; and separate phases. I am overall positive about the work and recommend the publication for Nature Communications, although some revisions and additions are needed before the work can be published. These recommendations are outlined below:

We highly value the positive feedback by the Reviewer. By addressing his/her constructive criticism, we were able to further strengthen the quality and impact of our study.

1. The authors discuss the relative contributions of neutral and charged hydrogen species, but the whole idea of radical species is not much supported by experimental methods, including for example EPR. Hence, the discussion remains for me a bit superficial and could be strengthened/further elaborated. The result of this is that the entire discussion on how the catalytic system actually operates; and what the effect of Ni is still vague; not clear to the reader/me. I have tried to envisage it but not completely understood it. So, there is more information/backing needed to explain what is exactly happening.

We agree that the presence of radicals is not conclusively proven by our experiments, but the well documented properties of metals to split hydrogen into neutral species, coupled to our DFT calculations and kinetic investigations, set a strong basis for this statement. Nevertheless, **we have attempted EPR measurements as suggested by the Reviewer**. Analyses of Ni-In₂O₃ with 0-10 wt.% Ni content have been carried out *in situ* at atmospheric pressure and 300 K after reduction in H₂ (10 mol.% in Ar) at 557 K. While intending to collect spectra at the reaction temperature (553 K), the signal was too weak under these conditions, necessitating a temperature reduction (**Figure 1**).

Figure 1. EPR spectra collected over 12 h for Ni-In₂O₃ catalysts with 1 and 5 wt.% Ni content and pure In₂O₃ serving as a reference after *in situ* activation in H₂ (10 mol.% in Ar) at 557 K, followed by cooling to 300 K in the same atmosphere.

In all cases, a sharp signal at *ca.* 4050 G is evident, which likely corresponds to localized oxygen defects, and is accompanied by a broad, weaker signal presumably associated with delocalized electrons, which become more abundant at higher nickel contents. In the 5 wt.% sample, a low-field

signal (3100 G) is additionally present, which could tentatively originate from antiferromagnetic NiO magnetically coupled to Ni³⁺ species. Still, the spectra are afflicted with a significant noise even after accumulating data for 12 h due to the conductive and ferromagnetic nature of the catalysts, prohibiting accurate data evaluation. These effects were substantially amplified for catalysts with higher (>5 wt.%) Ni contents, which could not be analyzed. Hence, the detection of hydrogen radicals by this technique remained elusive. While adequate protocols could potentially be developed in the future, we consider an investigation of the hydrogen speciation to such depth beyond the scope of this contribution. **Still, we extended the discussion on mechanistic aspects, integrating the role of hydrogen activation more clearly with our structural investigations, kinetic analyses, and insights by DFT (page 14, line 2 to page 15, line 3).**

2. The crux of the paper is the analysis of the XAFS spectra in combination with HRTEM; especially Figures 3b/c and the supporting information (Tables). Here, a more quantitative analysis, and extension to other catalyst materials are needed. Actually, I believe the paper would benefit much if the work was extended with a full analysis (including an expansion for Figure 6c to show next to the current samples also sample with 5 and 20 wt% Ni; hence where 5 samples are shown/discussed/analyzed in detail instead of the current 3 ones). Although I am aware that this requires more work it would much better focus on the different stages of Ni-In interaction. Now, some material is already some work included in the SI for the 5 wt% Ni sample, but more is clearly possible here, especially the PSI facility is close by. With this it would become more evident how an In-based methanol catalyst gradually changes in a Ni-based methane catalyst. Within this context I also refer to the use of IR probe molecules, which could differentiate between the alloying and different sites formed (see e.g. Nature Communications 2019, 10, 5330).

Thank you for this valuable comment. **We have extended the XAS and microscopy studies to the catalysts indicated and expanded Figure 3 accordingly**, which now follow the full structural evolution of Ni-In₂O₃ catalysts upon the addition of progressively higher amounts of nickel. **Additionally, we have broadened our kinetic investigations to the whole range of catalysts discussed in the manuscript and added the results to Figure 6 and new Supplementary Figures 12 and 13**, painting a much stronger picture of their structure-performance relationships. The additional experiments clearly show that reaction mechanism leading to CO or methanol are unaffected by the catalyst nanostructure, but that the rate at which H₂ is supplied to the RWGS path increases when larger alloy particles are formed. We also thank the Reviewer for the valuable suggestion regarding *operando* FTIR characterization. A successful application of the methodology outlined in the article mentioned would require the reactants (*i.e.*, CO₂) to adsorb on the nickel-based nanostructures. However, we have shown experimentally and through DFT simulations that this molecule barely adsorbs on the InNi₃ phase, as a key reason for the suppressed methanation behavior. Although acknowledging that other probe molecules could be relevant, we think that this demanding and, at the moment, explorative approach would deserve a stand-alone study. **Still, we have expended our thermal and CO-DRIFTS analyses (Figure 2) to embrace a broader range of nickel-promoted samples, as for XAS and microscopy. The insights of all these further investigations have been duly described in the amended manuscript (page 7, lines 6-22).**

REVIEWERS' COMMENTS

Reviewer #1 (Remarks to the Author):

The authors have properly addressed all my concerns and questions. I would recommend the manuscript to be published in the journal of Nature Communications.

Reviewer #3 (Remarks to the Author):

I have now read the revised version of the article, as well as the comments, including the comments which I have made earlier (referee #3). I am still convinced that this article deserves publication in Nature Communications, hence I recommend it for publication, after making the revisions. As a side note, I do not agree with the EPR assessment, although it is clear that this method will not deliver what I had hoped for. The authors refer to oxygen defects, and Ni phase, etc. I would have measured the entire field sweep range 0-7000 G; and put g values on the different signals, clearly for the 5Ni/In₂O₃ there are for me three signals; a signal, which could have a g value around 4 (guess I did not calculate it); would then be an impurity of Fe in the catalyst (not uncommon as EPR is so sensitive); and a broad signal which is most probably antiferromagnetic Ni oxide; and then a sharp although weak signal due to organic radical impurities; although for me a zoom-in could then differentiate between an oxygen defect site and this organic as I cannot judge the symmetry of the signal nor the exact line width, etc. Anyway, this would lead away from the main conclusions of the paper, and could be put in the supporting information if found interesting. Summarising, a very nice continuation of the ETH group, and other collaborators, in an important field of research. Hence, I strongly recommend it for publication in Nature Communications.

Manuscript NCOMMS-20-40209A – Response to Reviewers

Comments in *blue* – Replies in black

Reviewer #1

The authors have properly addressed all my concerns and questions. I would recommend the manuscript to be published in the journal of Nature Communications.

We are pleased that the Reviewer acknowledges the thoroughness of our revision and thank him/her for the recommendation to publish our work.

Reviewer #3

I have now read the revised version of the article, as well as the comments, including the comments which I have made earlier (referee #3). I am still convinced that this article deserves publication in Nature Communications, hence I recommend it for publication, after making the revisions. As a side note, I do not agree with the EPR assessment, although it is clear that this method will not deliver what I had hoped for. The authors refer to oxygen defects, and Ni phase, etc. I would have measured the entire field sweep range 0-7000 G; and put g values on the different signals, clearly for the 5Ni/In₂O₃ there are for me three signals; a signal, which could have a g value around 4 (guess I did not calculate it); would then be an impurity of Fe in the catalyst (not uncommon as EPR is so sensitive); and a broad signal which is most probably antiferromagnetic Ni oxide; and then a sharp although weak signal due to organic radical impurities; although for me a zoom-in could then differentiate between an oxygen defect site and this organic as I cannot judge the symmetry of the signal nor the exact line width, etc. Anyway, this would lead away from the main conclusions of the paper, and could be put in the supporting information if found interesting. Summarising, a very nice continuation of the ETH group, and other collaborators, in an important field of research. Hence, I strongly recommend it for publication in Nature Communications.

We appreciate the Reviewer's careful assessment of our amended manuscript and his/her highly positive feedback on its impact. We also value his/her insights into the EPR measurements conducted upon the revision. While the learning from this analysis does not further the depth of our global conclusions, as indicated by the Reviewer, we think that this technique holds potential to elucidate transversal properties of In₂O₃-based catalysts and plan a stand-alone investigation on its basis. Thus, we have refrained from adding these preliminary data to the Supplementary Information.